# Perceptions of risk in pregnancy with chronic disease: A systematic review and thematic synthesis

Elizabeth R. Ralston[1]*, Priscilla Smith[2], Joseph Chilcot[3], Sergio A. Silverio[1‡], Kate Bramham[1,2‡]

1 Department of Women and Children's Health, School of Life Course Sciences, King's College London, London, United Kingdom, 2 Department of Renal Medicine, King's Kidney Care Centre, King's College Hospital, National Health Service Foundation Trust, London, United Kingdom, 3 Department of Psychology, Institute of Psychiatry, Psychology and Neuroscience, King's College London, London, United Kingdom

‡ These authors are joint senior authors on this work.
* elizabeth.ralston@kcl.ac.uk

## Abstract

### Background

Women with chronic disease are at increased risk of adverse pregnancy outcomes. Pregnancies which pose higher risk, often require increased medical supervision and intervention. How women perceive their pregnancy risk and its impact on health behaviour is poorly understood. The aim of this systematic review of qualitative literature is to evaluate risk perceptions of pregnancy in women with chronic disease.

### Methods

Eleven electronic databases including grey literature were systematically searched for qualitative studies published in English which reported on pregnancy, risk perception and chronic disease. Full texts were reviewed by two researchers, independently. Quality was assessed using the Critical Appraisal Skills Programme Qualitative checklist and data were synthesised using a thematic synthesis approach. The analysis used all text under the findings or results section from each included paper as data. The protocol was registered with PROSPERO.

### Results

Eight studies were included in the review. Three themes with sub-themes were constructed from the analysis including: Information Synthesis (Sub-themes: Risk to Self and Risk to Baby), Psychosocial Factors (Sub-themes: Emotional Response, Self-efficacy, Healthcare Relationship), and Impact on Behaviour (Sub-themes: Perceived Risk and Objective Risk). Themes fitted within an overarching concept of Balancing Act. The themes together inter-relate to understand how women with chronic disease perceive their risk in pregnancy.

**Data Availability Statement:** All relevant data are within the manuscript and its Supporting Information files.

**Funding:** The authors received no specific funding for this work.

**Competing interests:** The authors have declared that no competing interests exist.

## Conclusions

Women's pregnancy-related behaviour and engagement with healthcare services appear to be influenced by their perception of pregnancy risk. Women with chronic disease have risk perceptions which are highly individualised. Assessment and communication of women's pregnancy risk should consider their own understanding and perception of risk. Different chronic diseases introduce diverse pregnancy risks and further research is needed to understand women's risk perceptions in specific chronic diseases.

## Introduction

Pregnant women who suffer from chronic disease such as epilepsy, kidney disease and diabetes are at increased risk of adverse pregnancy outcomes [1–4] and pregnancy can negatively impact chronic disease. For example, in chronic kidney disease pregnancy can accelerate decline in kidney function [4]. High-risk pregnancies often require greater medical supervision and care. However, increased objective risk does not always correlate with women's own risk perceptions [5, 6]; and many women with chronic disease do not perceive their pregnancy risk as severe [7, 8].

Risk is a subjective evaluation to help individuals understand the dangers and uncertainty in life [9, 10]. Two key dimensions to risk perception have been proposed: perceived susceptibility which is the likelihood of harm occurring and perceived severity which is the extent of harm caused [11]. Perceived susceptibility is a belief embedded within the Health Belief Model [12], which predicts preventative health behaviours through a set of core beliefs [13]. Others have conceptualised risk perception as the objective medical risk estimate and the subjective, socially constructed estimation of risk composed of social, psychological, and environmental factors [14–16].

A woman's perception of pregnancy risk affects her pregnancy-related decision making and behaviour, including delivery considerations and engagement with healthcare professionals (HCP) [14, 17–19] and may impact on pregnancy outcome.

Furthermore, a recent meta-synthesis reported women perceiving their pregnancy risk differently from their HCP [20]. However, understanding perception of pregnancy risk in women with chronic disease, in order to optimise communication about pregnancy risk is limited.

This review will contribute to the understanding of how women with chronic disease perceive their pregnancy risk. To the authors' knowledge, there is no comprehensive systematic thematic synthesis of qualitative studies exploring risk perceptions of pregnancy in women with chronic disease. An exploration of the existing qualitative research in pregnancy with chronic disease will add depth to existing knowledge and guide future research to improve pregnancy care. The aim of this systematic review of qualitative literature is to describe risk perceptions of pregnancy in women with chronic disease.

## Materials and methods

### Search strategy

The Preferred Reporting Items for Systematic Reviews and Meta Analyses (PRISMA;21) was followed and the pre-specified protocol was registered in PROSPERO (Registration number: CRD42019132367).

Eleven databases were searched between 1990 and 2020: MEDLINE, EMBASE, Global Health, PsychINFO, CINAHL, Scopus, Web of Science. Grey literature databases were also searched OpenGrey, British Library Electronic Theses Online Service, Dart European and OpenAIRE. The search was initially performed in April 2019 and updated in January and August 2020. The search strategy combined three key terms: 1: risk perception, and 2: pregnancy, and 3: chronic disease. Several chronic diseases were directly searched, and broad umbrella terms of chronic disease were otherwise searched. The search strategies for each database are presented in S1 Table. The selection process was guided by the four stages within the PRISMA statement: identification, screening, eligibility and inclusion [21]. Data were initially identified from the search and extracted, with duplicates removed. One author (ERR) conducted the extraction and preliminary screening of the titles and abstracts. Two authors (ERR & PS) independently screened full texts with the inclusion criteria. In all cases, disagreement was discussed and resolved.

## Inclusion criteria

Qualitative studies were included if published in English and included any data reporting perceptions of risk in women with chronic disease. Studies were included if participants were currently pregnant (irrespective to gestational age), previously pregnant or planning a pregnancy only if participants had a diagnosis of chronic disease prior to pregnancy. This included, but was not limited to: chronic kidney disease, chronic lung disease, chronic rheumatological disease, type 1 or type 2 diabetes mellitus, chronic hypertension, epilepsy, and/or coronary artery disease. Studies focusing on long-term mental health problems were excluded. Studies were eligible if they reported women's risk perceptions. Qualitative findings within mixed methods studies were included. Quantitative research, systematic reviews, meta-analyses, commentaries, and editorials were excluded.

## Study quality appraisal

Included studies were appraised using the Critical Appraisal Skills Programme Qualitative checklist which assesses methodological quality across ten items [22]. Item 10 was amended for reporting reasons from a free text response "how valuable is the research" to "is the research valuable". The checklist was chosen as the prompts provided between each item facilitates appraisal and reduces ambiguity of the question when there are multiple reviewers [23]. Two independent reviewers assessed the quality of the included studies (ERR & PS).

## Data analysis

It is important to define in qualitative systematic reviews what constitutes as data. Previous research has defined qualitative data as being the key concepts reported in the findings [24] or as all the text under the results or findings sections [25]. This analysis adopted the later approach and extracted all the text under the results or findings section. Data were analysed using NVivo 12 (version 12.6.0). Thematic synthesis was conducted following established guidelines [25]. Analysis consisted of three stages. First the results or findings section of each manuscript was line-by-line coded. Coding was inductive and guided by data. Second, codes were examined for similarities and differences then grouped, capturing the meaning of the initial codes. This developed descriptive themes. Third, descriptive themes were analysed further to generate analytical themes which reflected the synthesis of all included manuscripts. The coding and themes were reviewed independently by an author (PS) not involved in the coding process to check for suitability on two randomly selected papers.

# Results

## Description of studies

The search identified 4,024 citations, and 3,229 abstracts underwent screening after duplicates were removed. After screening 92 full texts against the eligibility criteria, 8 studies were included. The review flowchart is presented in Fig 1. Included studies focussed on either a single or combination of chronic diseases. These included chronic kidney disease [26], systemic lupus erythematosus [27], epilepsy [28, 29], diabetes mellitus [26, 30–32], congenital heart disease (CHD) [26, 33], human immunodeficiency virus [26], lupus/thyroid disease [26], genetic muscular disorder [26], cardiac valve repair [26] and/or ulcerative colitis [26]. Studies were conducted in five counties: United Kingdom (n = 3), Australia (n = 1), Brazil (n = 1), United States of America (n = 2), and Norway (n = 1). Characteristics of included studies are summarised in Table 1.

## Study quality

Both reviewers assessed the quality of the studies to be a high standard. A summary of the individual quality assessment is presented in S2 Table. The calculated percentage agreement for quality between both reviewers was 95%.

## Themes

Three main themes were generated from the analysis, each with sub-themes: (1) Information synthesis (including risk to self and risk to baby), (2) Psychosocial Factors (including emotional response, self-efficacy, healthcare relationship) and (3) Impact on Behaviour (including perceived risk and objective risk). Table 2 contains the overview of generated themes. Each theme is discussed in further detail below with supporting quotations.

## Theme 1: Information synthesis

The synthesis of risk information appeared to be compartmentalised into understanding risk to mother and risk to baby, as independent entities.

**Risk to self.**  Women actively seek information regarding risk to themselves about pregnancy and the impact on their chronic disease. Studies report healthcare professionals providing risk information in consultations and via leaflets [26–33]. Information from HCPs is sought to provide reassurance and positive feedback [26, 30]. There is a balance to be achieved in the quantity of information provided as too much was regarded as overwhelming and stressful [26, 32], whereas women felt uninformed when there was a lack of information [27, 28].

There was dissatisfaction with the lack of information provided [27, 28, 30–32]. This affected some women's trust in the HCP [31]. This was attributed to the limited time with HCPs and information not being available or accessible [28, 32]. Women describe adopting a proactive approach to seek information from peers, books and the internet [27–29]. Parous women reflected on their previous experiences to assess their pregnancy risk [29, 32]. Although for some women the desire to become a mother and experience pregnancy outweighs their concern about the risk to themselves [28, 33].

*"Their strong desire to have a child, similar to those of healthy women, indicated that the drive for motherhood was stronger than the drive for self-care."*

(Ngu et al., 2014; Congenital Heart Disease; Authors' Text)

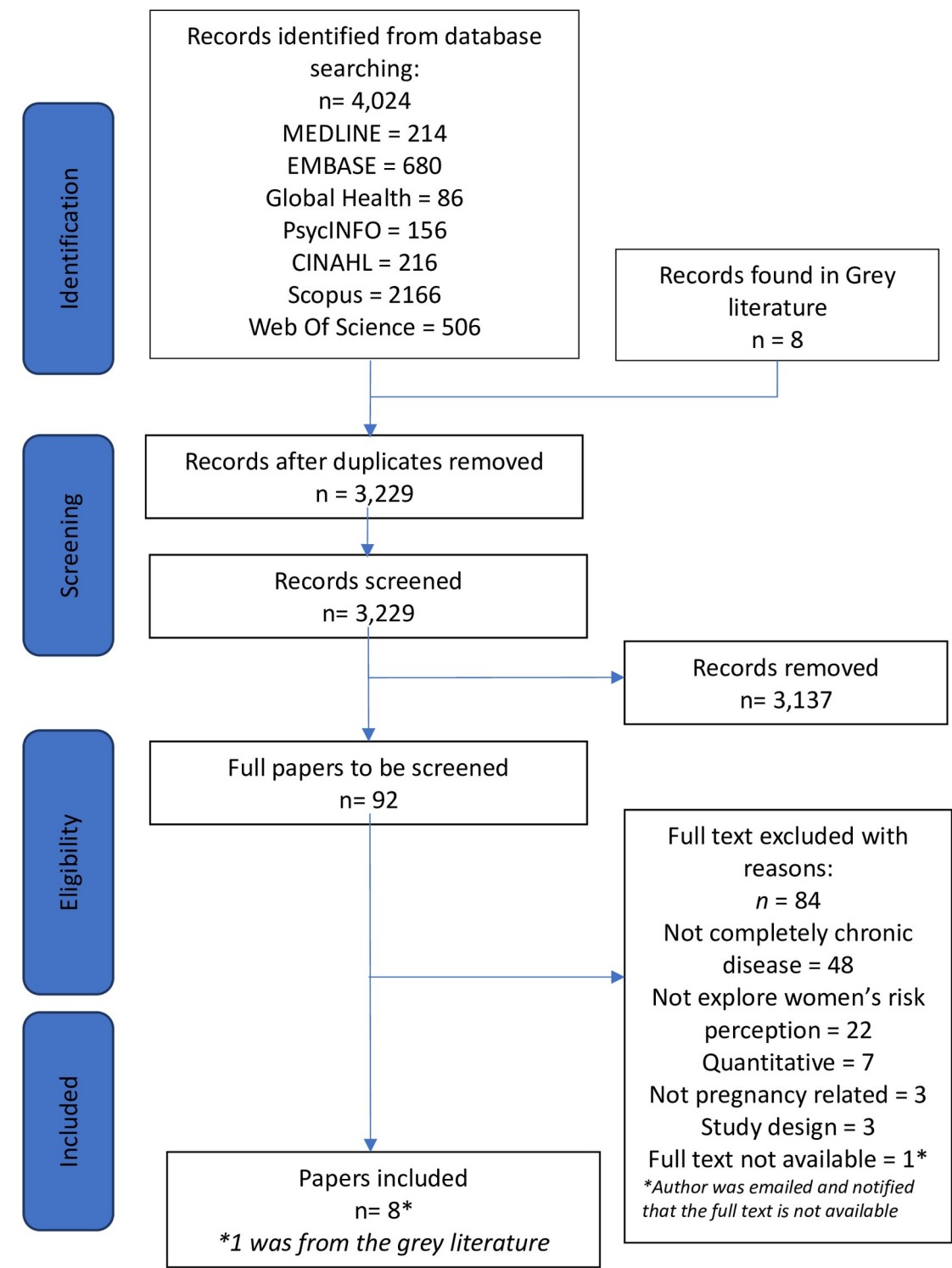

**Fig 1. PRISMA diagram showing identification and selection of studies.**

**Risk to baby.** There were concerns regarding the impact of chronic disease on the baby. Concerns included fetal macrosomia for women with diabetes [30, 31] and congenital malformations for women with prescribed medication for epilepsy. Women with systemic lupus

**Table 1. Summary of characteristics of included studies.**

| Author | Country | Research design | Sample (N) | Aims | Chronic Condition | Pregnancy inclusion criteria | Pregnancy status and previous pregnancy experience | Analysis |
|---|---|---|---|---|---|---|---|---|
| Boardman (2013) [28] [1] | United Kingdom | Qualitative semi-structured interview | 7 | 1) To understand how epilepsy impacts on the experience of pregnancy, labour and birth. 2) To understand how pregnancy impacts on the experience of epilsepy. | Epilepsy | Pregnant (≤29 weeks) or delivered within 9 months | At the time of interview, three women were between 23 weeks and 29 weeks gestation. Four women were between 2.5 and 8.5 months postpartum. It was the first pregnancy for six women. | Interpretative Phenomenological Analysis |
| McCorry et al (2012) [30] | United Kingdom | Qualitative semi-structured interview | 14 | 1) To explore attitudes toward pregnancy planning and preconception care seeking among women with diabetes. | Type 1 Diabetes mellitus | Preconception or postpartum | Four women had previous pregnancies. | Phenomenological Approach |
| Ngu, Hay, & Menahem (2014) [33] | Australia | Case studies, Qualitative semi-structured interview | 20 | 1) To understand the motivations of women with CHD to bear children. 2) To test the correlation of the clinicians and patients assessment of risk of the CHD to mother and child. 3) Assess if there are any discernable differences between the cohort with low risk CHD and those with high risk CHD. | Congenital heart disease | Completed 1 + successful pregnancies | Nine women had 1 child. Seven women had 2 children and four women had 3 children. | Thematic analysis |
| Rodrigues, Pereira Alves, Fialho Sim-Simc, and Surita (2020) [27] | Brazil | Qualitative Semi-structured interviews | 26 | To understand the meanings attributed to pregnancy by pregnant women with systemic lupus erythematosus during prenatal care. | Systemic Lupus Erythematosus | Currently pregnant | At the time of interviews all women were near 30 weeks gestation. Seven women had previously experienced miscarriages, 14 women had living children and five were in their first pregnancy. | Content analysis |
| Singh, Ingersoll, Gonder-Frederick, & Ritterband, (2019) [31] | United States | Qualitative semi-structured interviews | 15 | To explore women's experiences with and perceptions of pregnancy-related diabetes management and support systems that hinder or facilitate their self-management efforts. | Type 1 Diabetes mellitus | Pregnant, planning pregnancy or has experienced pregnancy | At the time of interview three women were currently pregnant. 10 women were previously pregnant, and five were planning a pregnancy. | Thematic analysis |

(*Continued*)

**Table 1.** (Continued)

| Author | Country | Research design | Sample (N) | Aims | Chronic Condition | Pregnancy inclusion criteria | Pregnancy status and previous pregnancy experience | Analysis |
|---|---|---|---|---|---|---|---|---|
| Tyer-Viola & Lopez (2014) [26] | United States | Qualitative semi-structured interview | 8 | To explore and describe the experience of pregnancy from the perspective of eight pregnant women with chronic illness | Chronic Kidney Disease, HIV, Lupus/Thyroid disease, Ulcerative Colitis, Genetic Muscular Disorder, Cardiac Valve Repair Type 1 Diabetes mellitus | Recruited when currently pregnant (>12 weeks, <24 weeks) | At the time of interview all women were past 36 weeks gestation or had delivered their baby within the last month. | Qualitative Description Analysis |
| Widnes, Schjøtt, & Granas (2012) [29] | Norway | Qualitative semi-structured interview | 10 | To examine risk perceptions and needs for medicines information in pregnant women with epilepsy | Epilepsy | Currently pregnant (>18 weeks) | At the time of interview women were between 20 to 34 weeks gestation. Four women had previous children. | Systematic text condensation |
| Wotherspoon, Young, McCance, & Holmes (2017) [32] | United Kingdom | Qualitative semi-structured interview | 11 | To provide insight into the knowledge of pre-eclampsia and views on implementation of a potential screening test for the condition in women with type 1 diabetes. | Type 1 Diabetes mellitus | Pre-conception, currently pregnant, up to 1 year post-partum | At the time of interview two women were planning a pregnancy. Nine women were currently pregnant with a mean gestation age of 24.6 weeks. | Thematic analysis |

[1]Thesis extracted from grey literature.

erythematosus and chronic hypertension were also concerned about the potential teratogenic impact of their medication [28, 29]. Although greater advances in healthcare and technology created the beliefs for some they would have a healthy outcome [33]. There was substantial concern for women about whether their baby would inherit their chronic disease, this potential inheritance was associated with feelings of guilt [26, 28, 31, 33].

> *"The majority of participants were also very anxious at the prospect of their babies being diagnosed with type 1 diabetes at birth or later in life. They expressed concern that they would feel very guilty if that were to happen and would 'wonder whether it was a selfish decision to have tried to become pregnant'."*

(Singh et al., 2019; Type 1 Diabetes Mellitus; Authors' Text)

**Table 2.** Overview of themes and sub-themes.

| Theme | Sub-theme |
|---|---|
| 1. Information synthesis | 1.1 Risk to self |
| | 1.2 Risk to baby |
| 2. Psychosocial factors | 2.1 Emotional response |
| | 2.2 Self efficacy |
| | 2.3 Healthcare relationship |
| 3. Impact on behaviour | 3.1 Perceived risk |
| | 3.2 Objective risk |

## Theme 2: Psychosocial factors

**Emotional response.**   Throughout the studies women expressed a level of pregnancy-related fear, anxiety and concern [26–33]. This heightened anxious response was exacerbated with the fear of harming the baby [26, 28, 30, 31]. The unknown pregnancy outcome further enhanced women's anxiety. Women describe living with chronic disease in itself as complex and stressful, pregnancy further added to this stress [26–28, 31, 32]. Pregnancy with chronic disease is emotionally complex and to cope women describe celebrating smaller pregnancy milestones to create a sense of normality [26].

*"Participants felt that they could not get too excited about their pregnancies because the reality was that they did not know what was going to happen to them, the pregnancies, or their infants, and things may not end up as planned"*

(Tyer-Viola & Lopez, 2014; Chronic Disease not Specified; Authors' Text)

**Self-efficacy.**   Women describe the difficulty managing and maintaining their chronic disease prior to pregnancy [26, 28, 30, 31]. These management challenges experienced prior to pregnancy fostered a lack of self-belief in women being able to cope with pregnancy as well as their chronic disease [28, 30, 31] as pregnancy created an additional level of complexity. The unpredictable nature of chronic disease was highlighted, particularly for women with type 1 diabetes, who were concerned what impact fluctuating glucose levels would have on their baby [30, 31]. Self-doubt was also described postpartum as women described concerns regarding their ability to care for the baby and fulfil motherhood responsibilities [27, 28].

"*How am I going to take care of a baby if I can't even take care of myself properly? It has to be bathed, taken care of. . . A baby needs all sorts of care, so I ask myself 'how are you going to do this? On days you are in pain, how are you going to handle it? On days that you are sick, with whom will you leave the baby [with] so you can go to the hospital?'*"

(Rodrigues et al., 2020; Systemic Lupus Erythematosus; Participant's Quotation)

**Healthcare relationship.**   Throughout the studies the importance of a positive relationship with good communication between women and their HCP was demonstrated [26–33]. One study reported women's trust was enhanced when their HCP was engaged with up-to-date research [29]. The relationship was a source of trust and guidance. The increase in interactions between women and their HCP throughout their pregnancy was a source of reassurance to women [28, 30]. When this relationship was not developed, women reflected adversely upon their interactions with their HCP [28, 30, 31]. Components of this negative reflection include a lack of interpersonal skills, being too medicalised and lacking a holistic approach [28]. However, a medicalised approach was accepted if women perceived it to be the safest option for the baby [27, 28].

"My CDE (certified diabetes educator) is wonderful because she is like my cheerleader. She is trying to get me through my goals. . . and even though things aren't always the greatest, she will point out the positive things. . ."

(Singh et al., 2019; Type 1 Diabetes Mellitus; Participant's Quotation)

## Theme 3: Impact on behaviour

**Perceived risk.**   This sub-theme describes women adjusting their behaviour to reduce the risk to themselves and their baby which is a consequence of how women perceive their pregnancy risk. Studies describe women changing their behaviour as a result of an increased awareness of their pregnancy risk. These changes include improving medication adherence and following medical advice more stringently [27–32]. Women discussed increasing their engagement with their HCP to benefit their pregnancy [27]. There was a shift from having a relatively relaxed approach when managing chronic disease to becoming risk averse. Several women perceived a greater sense of responsibility over their pregnancy due to having a chronic disease [28, 30, 31]. Thus, women describe increasing the monitoring of their chronic disease, such as increasing the frequency of blood glucose assessment for women with diabetes [30, 31]. Some women describe becoming more involved in their pregnancy-related decision making and adopting a proactive role in their pregnancy care.

*"I take it every day now whereas before I was pregnant I was a bit lax with it"*

(Boardman, 2013; Epilepsy; Particiapnt's Quotation)

**Objective risk.**   This sub-theme describes the impact objective risk information has on pregnancy-related behaviour when information is taken more literally. It captures the influence of providing risk information without accounting for individual psychosocial factors. One study described the provision of objective risk information leading to a change in decisions around pregnancy [31]. One woman described how the information around potential pregnancy complications discouraged them from future pregnancy [30]. Boardman et al. [28] reported several HCPs expecting women to follow their instructions based on risk information provided. There was an expectation by HCPs that women would adhere to medical advice due to their objective risk. It is considered that women following their HCP stringently may place greater responsibility upon their HCP [33].

*"My primary care physician flat out told me, 'You should not get pregnant. It's going to be miserable and hard with your blood sugars.' I said, 'thanks, but I want to be a mom!' Last summer, I started looking into adoption because they said I might never become pregnant."*

(Singh et al., 2019; Type 1 Diabetes Mellitus; Participant's Quotation)

## Interpretation of relationship

The themes identified all relate to one another by an overarching concept of Balancing Act. Women with chronic disease were attempting to maintain a balance across several components of their health and pregnancy-related decision making, including synthesising information, too little information was considered as uninformed, but much was overwhelming [26–28, 30, 32]. Women sought information which was reassuring, but would balance this with the reality of the associated risks of pregnancy with chronic disease. Women described their emotions and anxiety over the unknown pregnancy outcomes [26, 27, 30] and would balance these emotions with creating small, attainable milestones to celebrate their progress [26]. When women discussed the possibility of trying to conceive, their decision was balanced with weighing potential pregnancy risk to both mother and baby [30]. This concept is captured by Tyer-Viola and Lopez [26] who describe 'balancing between the fantasy versus the reality'.

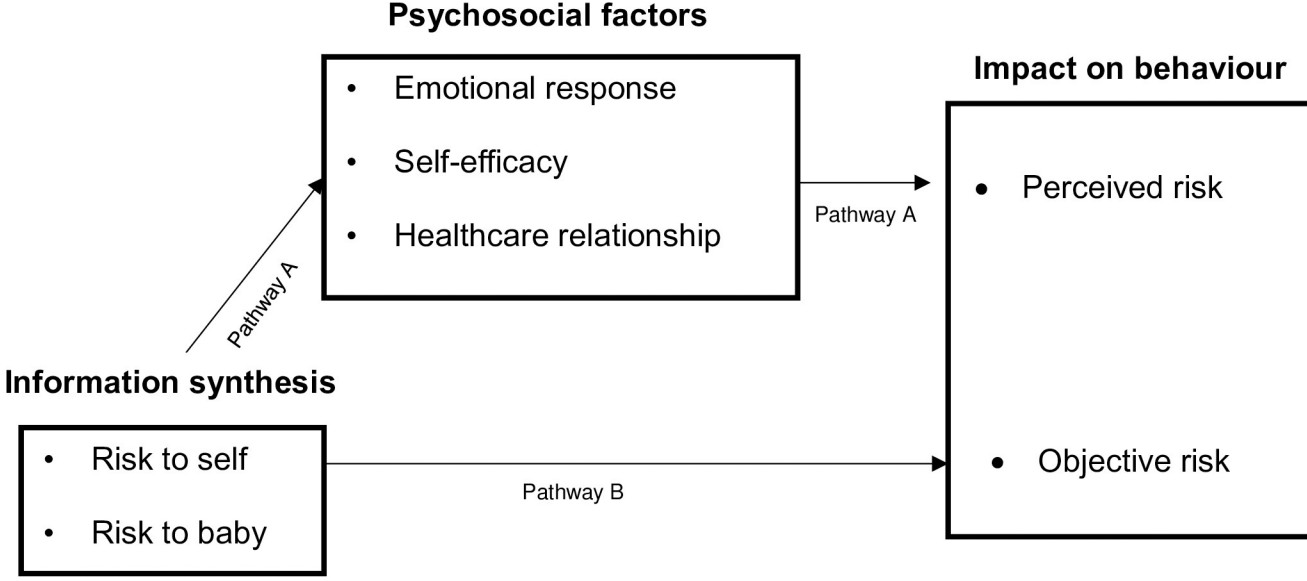

**Fig 2. Proposed pathway of themes.**

**Inter-relationship between themes.** The themes generated from this synthesis inter-relate to understand how women with chronic disease perceive their risk in pregnancy. Fig 2 illustrates proposed pathways to describe the inter-relationship between themes. Pathway A is supported by the most compelling data from the synthesis integrating all three themes together. This pathway suggests women initially synthesise information regarding pregnancy risk to themselves and their baby which is influenced by women's individual psycho-social factors to form women's perception of risk, which impacts women's pregnancy-related behaviour. Pathway B does not account for psychosocial factors and suggests pregnancy-related behaviour is the response to objective risk information reflecting a traditional medical model. The proposed inter-relationships between themes suggest women's pregnancy-related behaviour is often impacted by their perceived risk, which is formulated from the synthesis of risk information and psychosocial factors thus the data analysis supports Pathway A.

## Discussion

The aim of this systematic review of was to describe risk perceptions of pregnancy in women with chronic disease and its impact on behaviour. Three themes were generated from the analysis: Information synthesis; Psychosocial Factors; and Impact on Behaviour. Conceptually together they can be described as a Balancing Act which women perform when they are pregnant with a chronic disease.

Information synthesis highlighted two key messages: a lack of information being provided to women and compartmentalisation of women understanding risk to themselves and the risks to the baby. These findings are aligned with other reports of women's frustration regarding inadequate presentation of maternal and neonatal risk, and requirements for evidence regarding risk and improved communication for pregnant women with chronic disease [34, 35]. Women also desired more information about medication, as they had concerns regarding teratogenicity. These concerns and negative beliefs could contribute to reduced adherence, as the association between negative medication beliefs and low medication adherence in pregnancy has been identified [36].

In keeping with studies assessing risk perceptions in low-risk pregnancies, parous women reflect upon previous pregnancy experiences to assess their current pregnancy risk [37, 38], reflecting use of the availability heuristic [39]. This reinforces the importance of information provision to allow women to develop heuristics to assist their decision-making process.

An important consideration proposed by Relph *et al.* [40] who described that discussions around pregnancy risk between women with obesity and their healthcare provider appeared to be avoided unless risk arose, suggesting there may be reluctance from both HCPs and women to initiate discussion about risk. It may be that HCPs are concerned of introducing unwarranted anxiety over potential risks that may not occur [40], although Garrud, Wood, and Stainsby [41] reported that providing detailed risk information was associated with greater knowledge with no increase in anxiety. Limited information provided may give women a false sense of security in their perception of pregnancy risk.

The studies included in the systematic review highlighted that women view risk separately for themselves and their baby, supporting the notion that the fetus is its own entity [42] and that two distinct risk appraisals are undertaken by women [43, 44]. This separation of risk may be an indication of women creating a hierarchy of importance with priorisiation of their baby. Research has reported a shift in focus from the mother to the baby, with the baby's safety being paramount [35, 43, 45]. This has led to women feeling overlooked [40] and so it is important women's own needs are not diminished by the baby's needs.

Several studies included in the analysis reported that individual factors contribute to perception of pregnancy risk for women with chronic disease. Previous studies have reported heightened maternal anxiety in high-risk pregnancies [5, 7, 37, 46], and associations are described between raised anxiety and women perceiving they have increased pregnancy risk [7]. Increased emotional responses described in the studies may indicate women with chronic disease perceive a greater pregnancy risk. In support, perceived low self-efficacy during pregnancy and postpartum may indicate a higher perception of pregnancy risk which has been described by others [47]. In addition, perceptions of lack of control and inadequacy contributed to heightened emotional responses in women with chronic disease, including stress and anxiety. It is likely to be important to focus on improving women's self-efficacy around managing their chronic disease during pregnancy including postpartum to help to alleviate their heightened stress and anxiety.

The relationship between women and their HCP also influences women's perception of pregnancy risk and their satisfaction with care. Women frequently sought HCP opinions and psychosocial support when discussing pregnancy risk [37, 38, 48, 49], this was associated with a perceived responsibility shift, either with greater responsibility placed upon the woman or their HCP. Other studies of high-risk pregnancies, have also described women adopting greater responsibility for their baby [46] or accepting either an active or passive role toward their pregnancy-related decision-making [48].

The synthesis also suggests women's behaviour is impacted by their perception of risk and their objective risk. Women's behavioural adjustments to reduce their perceived pregnancy risk included improved engagement with their healthcare provider as described by Rodrigues et al. [27] and greater adherence to medical advice as Boardman [28] reported. This suggests the degree to which women adhere to medical advice may be related to their level of perceived pregnancy risk [44]. This is supported by the Health Belief Model which proposes that an individuals perceived susceptibility of risk can predict preventative behaviour [12, 13], providing rationale why women's behaviour is impacted by their perception of pregnancy risk.

In addition, the analysis highlighted that objective risk also impacted on women's behaviour although appeared to be less important than perceived risk. Some women changed their behaviour based on the risk information provided. HCPs expected women to follow their

medical advice and often assumed that women perceive their risk according to their objective risk. However, several studies reported that women and HCPs do not perceive pregnancy risk in the same way [20, 50–52], although some women may feel they have little power to question medical authority [53].

The pathways constructed in this review suggest women's pregnancy-related behaviour is impacted by how they perceive their pregnancy risk. Overall, the findings highlight it is important for women to form an accurate perception of their pregnancy risk, concordant with their objective risk, so they are able to make informed decisions regarding their pregnancy care. It is important to remember that in pregnancy women with chronic disease are constantly trying to maintain a balance between fear and excitement. Risk perception is unique to each individual and not solely based on objective risk information, and so an assessment of women's pregnancy risk should consider their own understanding and perception of risk.

## Limitations

The limitations of this review are firstly the small number of studies which met eligibility criteria, highlighting the paucity of relevant research. There was substantial heterogeneity between studies, with diverse overall aims, and inclusion of women at different gestations as well as pre-conception, with different chronic diseases. However, there were some consistencies across studies, supporting common factors influencing pregnancy risk perception and behaviour in women with chronic disease. In reagrds to study quality, Ngu, Hay and Menahem [33] and Singh et al. [31] did not address the relationship between the researcher and their participants. This is an important reflection as the researcher is an integral part of the research, which introduces power imbalance and bias [54].

## Implications for research and practice

The findings of this analysis suggested that resources providing information regarding risk need to be improved to allow understanding of objective risk. Relationships with HCPs should be a source of reassurance and provide positive feedback to increase women's trust and confidence during pregnancy, and the importance of this relationship should be recognised and developed. HCPs should engage in conversations with women exploring self-belief and perceived control over caring for their baby during pregnancy and postpartum whilst having a chronic disease, which may help to reduce the heightened anxiety and stress. When providing risk information HCPs should consider women's psychosocial factors and explore their understanding of risk, as this will mediate the relationship between providing risk information and future impact on behaviour. Further research is needed to understand women's risk perceptions in specific chronic diseases; however this review highlights the individualised nature of risk perception and the impact of information synthesis and psychosocial factors upon pregnancy-related behaviour for women with chronic disease.

## Supporting information

**S1 Checklist. PRISMA 2009 checklist.**
(DOC)

**S1 Table. Search strategies for each database.**
(DOCX)

**S2 Table. Quality assessment of included qualitative papers using the Critical Appraisal Skills Programme Qualitative checklist (2018).**
(DOCX)

**S1 Record. International prospective register of systematic reviews.**
(PDF)

## Author Contributions

**Conceptualization:** Elizabeth R. Ralston, Priscilla Smith, Joseph Chilcot, Sergio A. Silverio, Kate Bramham.

**Data curation:** Elizabeth R. Ralston, Priscilla Smith.

**Formal analysis:** Elizabeth R. Ralston, Priscilla Smith, Sergio A. Silverio.

**Investigation:** Elizabeth R. Ralston.

**Methodology:** Elizabeth R. Ralston, Sergio A. Silverio, Kate Bramham.

**Project administration:** Elizabeth R. Ralston.

**Software:** Elizabeth R. Ralston.

**Supervision:** Joseph Chilcot, Sergio A. Silverio, Kate Bramham.

**Validation:** Elizabeth R. Ralston, Priscilla Smith, Joseph Chilcot, Sergio A. Silverio, Kate Bramham.

**Visualization:** Elizabeth R. Ralston, Sergio A. Silverio, Kate Bramham.

**Writing – original draft:** Elizabeth R. Ralston.

**Writing – review & editing:** Elizabeth R. Ralston, Priscilla Smith, Joseph Chilcot, Sergio A. Silverio, Kate Bramham.

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
