## [Decision Letter · Decision Letter 0]

2 Jun 2021

PONE-D-21-12200

Perceptions of risk in pregnancy with chronic disease: a systematic review and thematic synthesis

PLOS ONE

Dear Dr. Ralston,

Thank you for submitting your manuscript to PLOS ONE. After careful consideration, we feel that it has merit but does not fully meet PLOS ONE’s publication criteria as it currently stands. Therefore, we invite you to submit a revised version of the manuscript that addresses the points raised during the review process.

We look forward to receiving your revised manuscript.

Kind regards,

Xu-jie Zhou, Ph.D., M.D.,

Academic Editor

PLOS ONE

Journal Requirements:

2.Thank you for stating the following in the Acknowledgments Section of your manuscript:

"Sergio A. Silverio (King’s College London) is currently supported by the National

446 Institute for Health Research Applied Research Collaboration South London [NIHR

447 ARC South London] at King’s College Hospital NHS Foundation Trust. The views

448 expressed are those of the authors and not necessarily those of the NIHR or the

449 Department of Health and Social Care."

Reviewers' comments:

Reviewer's Responses to Questions

**Comments to the Author**

1. Is the manuscript technically sound, and do the data support the conclusions?

Reviewer #1: Yes

Reviewer #2: Yes

2. Has the statistical analysis been performed appropriately and rigorously? 

Reviewer #1: I Don't Know

Reviewer #2: N/A

3. Have the authors made all data underlying the findings in their manuscript fully available?

Reviewer #1: Yes

Reviewer #2: Yes

4. Is the manuscript presented in an intelligible fashion and written in standard English?

Reviewer #1: Yes

Reviewer #2: Yes

5. Review Comments to the Author

Reviewer #1: This is a major revision. The revisions address the comments raised before and improve the manuscript which reports novel findings. The manuscript still needs some attention as follows:

1. Please check the main text (including abstract), especially the references (e.g.: references 26, 37, 40, 47, 48), according to the journal submission guidelines. Article sections are defined as Introduction, Materials and Methods, Results, Discussion, and Conclusions (optional). It should be check according to this order and the journal submission guidelines.

2. Abstract, methods: More detailed information can be given.

3. Methods: Add outcome measurements.

4. Methods, inclusion criteria: Are mixed methods included, excluding quantitative results from these studies? If mixed methods are excluded from the screened articles, check and add them to the method and flow diagram.

5. Line 115: "Quantitative research, systematic reviews, and meta-analyses were excluded". Commentaries, editorials, and case reports/studies were encountered during screened? If available and excluded/included, please check and add to the inclusion criteria section.

6. Table 1: " Ngu, Hay and Menahem (2014), Research design: Qualitative semistructured interview". Isn't it necessary to also add the case studies part to the research design?

7. Study quality, Line 151: "Both reviewers assessed the quality of the studies to be a high standard. " Ngu, Hay and Menahem (2014) article is really high quality?

8. Table 1: Add aims and generally more detailed information can be given and needs rewriting for clarity.

9. Table 1, Pregnancy inclusion criteria: More detailed information can be given and needs.

10. Table 1, Tyer-Viola & Lopez (2014), chronic condition: Please check. If you make changes in the table, add changes in the description of study section.

11. Table 1, first line: Author: -> Author (year) [reference number]

12. Discussions: The articles included in the systematic review were rarely mentioned.

13. Limitations: It should also be added that some studies were conducted with pre-pregnancy women, not with pregnant women.

14. Line 72: [HCP] -> (HCP)

15. Line 98: AND -> and

16. Line 142: Table one -> Table 1

17. Line 169: remove " [HCP] "

Reviewer #2: Women with chronic disease are at increased risk of adverse pregnancy outcomes. Pregnancies which pose higher risk, often require increased medical supervision and intervention. How women perceive their pregnancy risk and its impact on health behavior is poorly understood. In this study, the authors conducted a systematic review to evaluate risk perceptions of pregnancy in women with chronic disease and found that: 1. women’s pregnancy-related behavior and engagement with healthcare services are influenced by their perception of pregnancy risk, 2. Women with chronic disease have risk perceptions which are highly individualized, assessment and communication of women’s pregnancy risk should consider their own understanding and perception of risk. The study design and method are suitable and the data could support the conclusions.

6. PLOS authors have the option to publish the peer review history of their article (what does this mean?). If published, this will include your full peer review and any attached files.

Reviewer #1: No

Reviewer #2: No

---

## [Author Response · Author response to Decision Letter 0]

1 Jul 2021

Reviewer 1:

1. Please check the main text (including abstract), especially the references (e.g.: references 26, 37, 40, 47, 48), according to the journal submission guidelines. Article sections are defined as Introduction, Materials and Methods, Results, Discussion, and Conclusions (optional). It should be check according to this order and the journal submission guidelines. 

Thank you for highlighting the formatting. The references have been corrected and updated. The article sections are now defined as; Introduction, Materials and Methods, Results and Discussion with sub-headings kept to three levels. 

2. Abstract, methods: More detailed information can be given.

We have added further detail into the abstract about the quality appraisal, analysis, and the inter-relationship between the themes. 

Abstract - Line 31-34 “Quality was assessed using the Critical Appraisal Skills Programme Qualitative checklist and data were synthesised using a thematic synthesis approach. The analysis used all text under the findings or results section from each included paper as data.” 

Abstract - Line 41 and 42 “The themes together inter-relate to understand how women with chronic disease perceive their risk in pregnancy.” 

Methods - Line 108-113):

“The selection process was guided by the four stages within the PRISMA statement: identification, screening, eligibility and inclusion [21]. Data were initially identified from the search and extracted, with duplicates removed. One author (ER) conducted the extraction and preliminary screening of the titles and abstracts. Two authors (ER & PS) independently screened full texts with the inclusion criteria.” 

Line 142-146: please see comment 3 below. 

Line 156-157: “The coding and themes were reviewed independently by an author (PS) not involved in the coding process to check for suitability on two randomly selected papers.” 

3. Methods: Add outcome measurements.

Thank you for your suggestion. We felt that with qualitative syntheses, outcome measurements – i.e. themes - are not easily defined prior to analysis. However, we recognise that it is important to define the difference between data being analysed and subsequent outcomes – themes in qualitative syntheses. In this review we defined data as being the text under the ‘results’ or ‘findings’ heading of each paper included. We have added this detail in below the sub-heading ‘Data analysis’ (Line 142-146). We are happy to adapt this further according to the editor’s wishes.

Line 142 – 146: “It is important to define in qualitative systematic reviews what constitutes as data. Previous research has defined qualitative data as being the key concepts reported in the findings [24] or as all the text under the results or findings sections. [25]. This analysis adopted the later approach and extracted all the text under the results or findings sections.” 

4. Methods, inclusion criteria: Are mixed methods included, excluding quantitative results from these studies? If mixed methods are excluded from the screened articles, check and add them to the method and flow diagram.

Thank you for highlighting this issue, qualitative findings of mixed methods studies were included. This has been clarified in the inclusion criteria: “Qualitative findings within mixed methods studies were included. Quantitative research, systematic reviews, meta-analyses, commentaries, and editorials were excluded.” (Line 128-130)

5. Line 115: "Quantitative research, systematic reviews, and meta-analyses were excluded". Commentaries, editorials, and case reports/studies were encountered during screened? If available and excluded/included, please check and add to the inclusion criteria section.

Thank you for identifying this concern. Commentaries and editorials were excluded from the review, we have added this into the inclusion criteria section (Line 129-130), please see updated quote in point 4 above. 

6. Table 1: " Ngu, Hay and Menahem (2014), Research design: Qualitative semistructured interview". Isn't it necessary to also add the case studies part to the research design?

Thank you for your comment. We have now added case studies into the research design in Table 1. Ngu, Hay and Menahem (2014) conducted 20 semi-structured interviews. And there were no other case studies identified in the manuscript reviews. 

7. Study quality, Line 151: "Both reviewers assessed the quality of the studies to be a high standard. " Ngu, Hay and Menahem (2014) article is really high quality?

We found the overall quality of the included papers to be of a high standard. In the Ngu, Hay and Menahem (2014) article according to the CASP checklist we found item 4 and 6 difficult to determine and item 8 as not sufficient. In Item 4 (“Was the recruitment strategy appropriate to the aims of the research?”) we found the recruitment strategy was not clearly defined and justified. In item 6 (“Has the relationship between researcher and participants been adequately considered?”), it was not possible to tell whether they had considered the impact of the relationship. We found that other papers also did not adequately address this and therefore, we included this in the Limitation section at line 469-472. Item 8 (“Was the data analysis sufficiently rigorous?”), we thought that further detail was required in the analysis especially as only one researcher conducted the thematic analysis and did not comment on their own bias/influence upon the analysis. We did rate Ngu, Hay and Menahem’s manuscript as the weakest study; however, overall the papers were to a high standard.

8. Table 1: Add aims and generally more detailed information can be given and needs rewriting for clarity. 

Thank you for your suggestion. We have added to Table 1 the aims of each paper, along with further information about participants pregnancy status and previous pregnancy experience. We would be happy to receive journal specific guidance to improve the table.

9. Table 1, Pregnancy inclusion criteria: More detailed information can be given and needs.

We have added a separate column with details of the pregnancy status at the time of participation and the previous pregnancy experience of the participants of the individual studies. We hope this has provided more clarity over the pregnancy inclusion criteria. 

10. Table 1, Tyer-Viola & Lopez (2014), chronic condition: Please check. If you make changes in the table, add changes in the description of study section. 

We have checked the chronic conditions included in Tyer-Viola & Lopez, and included the Lupus/Thyroid disease and cardiac valve repair. This has been added to Table 1 and the description of studies.

11. Table 1, first line: Author: -> Author (year) [reference number]

This has been updated in Table 1.

12. Discussions: The articles included in the systematic review were rarely mentioned.

Thank you for this comment. We have attempted to include references to manuscripts more frequently. 

13. Limitations: It should also be added that some studies were conducted with pre-pregnancy women, not with pregnant women.

Thank you for highlighting this. This has been added into the limitations that the studies included different gestations as well as preconception (Line 466).

Line 466: “There was substantial heterogeneity between studies, with diverse overall aims, and inclusion of women at different gestations as well as preconception, with different chronic diseases”

14. Line 72: [HCP] -> (HCP) 

Thank you for highlighting this, it has now been corrected.

15. Line 98: AND -> and

Thank you for highlighting this, it has now been corrected.

16. Line 142: Table one -> Table 1

Thank you for highlighting this, it has now been corrected.

17. Line 169: remove " [HCP] "

Thank you for highlighting this, it has now been removed.

Reviewer 2: Women with chronic disease are at increased risk of adverse pregnancy outcomes. Pregnancies which pose higher risk, often require increased medical supervision and intervention. How women perceive their pregnancy risk and its impact on health behavior is poorly understood. In this study, the authors conducted a systematic review to evaluate risk perceptions of pregnancy in women with chronic disease and found that: 1. women’s pregnancy-related behavior and engagement with healthcare services are influenced by their perception of pregnancy risk, 2. Women with chronic disease have risk perceptions which are highly individualized, assessment and communication of women’s pregnancy risk should consider their own understanding and perception of risk. The study design and method are suitable and the data could support the conclusions.

Thank you for your kind comment.

---

## [Decision Letter · Decision Letter 1]

7 Jul 2021

Perceptions of risk in pregnancy with chronic disease: a systematic review and thematic synthesis

PONE-D-21-12200R1

Dear Dr. Ralston,

We’re pleased to inform you that your manuscript has been judged scientifically suitable for publication and will be formally accepted for publication once it meets all outstanding technical requirements.

Kind regards,

Xu-jie Zhou, Ph.D., M.D.,

Academic Editor

PLOS ONE

Additional Editor Comments (optional):

Reviewers' comments:

Reviewer's Responses to Questions

**Comments to the Author**

1. If the authors have adequately addressed your comments raised in a previous round of review and you feel that this manuscript is now acceptable for publication, you may indicate that here to bypass the “Comments to the Author” section, enter your conflict of interest statement in the “Confidential to Editor” section, and submit your "Accept" recommendation.

Reviewer #1: (No Response)

Reviewer #2: All comments have been addressed

2. Is the manuscript technically sound, and do the data support the conclusions?

Reviewer #1: (No Response)

Reviewer #2: Yes

3. Has the statistical analysis been performed appropriately and rigorously? 

Reviewer #1: (No Response)

Reviewer #2: N/A

4. Have the authors made all data underlying the findings in their manuscript fully available?

Reviewer #1: (No Response)

Reviewer #2: Yes

5. Is the manuscript presented in an intelligible fashion and written in standard English?

Reviewer #1: (No Response)

Reviewer #2: Yes

6. Review Comments to the Author

Reviewer #1: (No Response)

Reviewer #2: (No Response)

7. PLOS authors have the option to publish the peer review history of their article (what does this mean?). If published, this will include your full peer review and any attached files.

Reviewer #1: No

Reviewer #2: No

---

## [Editor Report · Acceptance letter]

9 Jul 2021

PONE-D-21-12200R1 

Perceptions of risk in pregnancy with chronic disease: A systematic review and thematic synthesis 

Dear Dr. Ralston:

I'm pleased to inform you that your manuscript has been deemed suitable for publication in PLOS ONE. Congratulations! Your manuscript is now with our production department. 

Kind regards, 

on behalf of

Dr. Xu-jie Zhou 

Academic Editor

PLOS ONE